# Ethical and psychosocial considerations for hospital personnel in the Covid-19 crisis: Moral injury and resilience

**Alexander Kreh** [1]*, **Rachele Brancaleoni**[2], **Sabina Chiara Magalini**[3], **Daniela Pia Rosaria Chieffo**[2], **Barbara Flad**[4], **Nils Ellebrecht**[5], **Barbara Juen**[1]

**1** University of Innsbruck, Innsbruck, Austria, **2** Fondazione Policlinico Universitario A. Gemelli IRCCS, Rome, Italy, **3** Catholic University of the Sacred Heart, Rome, Italy, **4** A.ö. Krankenhaus St. Vinzenz, Zams, Austria, **5** Albert-Ludwigs-University, Freiburg, Germany

☯ These authors contributed equally to this work.
* alexander.kreh@uibk.ac.at

**Data Availability Statement:** Full data connot be shared publicly because they contain potentially identifiying or sensitive participant information.

## Abstract

This study aims at investigating the nature of resilience and stress experience of health care workers during the COVID-19 pandemic. Thirteen healthcare workers from Italian and Austrian hospitals specifically dealing with COVID-19 patients during the first phase of the pandemic were interviewed. Data was analysed using grounded theory methodology. Psychosocial effects on stress experience, stressors and resilience factors were identified. We generated three hypotheses. Hypothesis one is that moral distress and moral injury are main stressors experienced by healthcare workers. Hypothesis two states that organisational resilience plays an important part in how healthcare workers experience the crisis. Organisational justice and decentralized decision making are essential elements of staff wellbeing. Hypothesis three refers to effective psychosocial support: Basic on scene psychosocial support based on the Hobfoll principles given by trusted and well-known mental health professionals and peers in an integrated approach works best during the pandemic.

## Introduction

### Psychosocial aspects in the COVID-19 response

The COVID-19 pandemic has spread rapidly and has since become a severe mental health crisis. Measures aiming at the containment of the virus, protection of the risk population as well as mitigation changed the life and routines of many around the globe. Measures required during the response, such as quarantine, can have a severe impact on the mental health state of the population [1].

### Higher stress in healthcare workers during pandemics

By caring for patients infected by COVID-19, health care professionals all around the world are involved in the acute response to the crisis. Based on the experiences made during the

Excerpts of the transcripts relevant to the study are made available within the paper. Paraphrased, anonymized and conceptualized data after using grounded theory methodology can be retrieved from doi.org/10.5281/zenodo.4241858 The excerpts and conceptualized data are sufficient for independent researchers to verify the findings of this study.

**Funding:** This project has received funding from the European Union's Horizon 2020 research and innovation programme under grant agreement No. 786670. The funders had no role in study design, data collection and analysis, decision to publish, or preparation of the manuscript.

**Competing interests:** The authors have declared that no competing interests exist.

COVID-19 pandemic, the high exposure to stressors as well as the need for Psychosocial Support for health care workers has been well communicated [2–4]. According to findings from the SARS outbreaks in 2003, H1N1 outbreak in 2009, as well as after several outbreaks of Ebola, healthcare personnel working on the front-line might be at risk of experiencing higher levels of stress immediately after the response. Working on the front-line in pandemics can also negatively impact the development of mental health problems in the long-term which can include higher levels of PTSD symptoms, burnout and other mental health problems [5–14].

Similar results can now be found in COVID-19 [15–19]. Tan et al. [19] analysed data from 1257 Healthcare workers in 34 different hospitals from January 29th to February 3rd, 2020. Participants were aged between 26 und 40, 764 (60,8%) were nurses, 493 (39,2%) doctors. Many showed heightened signs of depression (50,4%), anxiety (44,6%), insomnia (34%) and feelings of distress (71,5%). Pappa et al. [18] did a meta analysis and found increased values for depression (23,2%) and anxiety (22,8%). Especially sex and type of job (women and nurses versus men and doctors) showed more affective symptoms. Insomnia was rated 38,9%.

## Stressors

Several factors that caused stress among health care professionals during earlier pandemics have been identified. Many of those stressors could also be verified in first studies on the COVID-19 outbreak. Dai et al. [20] did a study on concerns of 4357 healthcare workers. According to the data, their main concerns were the infection of colleagues (72,5%), infection of family members (63,9%), protection measures (52,3%) and ethical concerns (48,5%). 39,1% of the participants showed clinically relevant psychological problems, especially those who were based in Wuhan or those who were in quarantine and/or had infected a colleague or family member.

## High exposure

Generally, concerns about infecting family members and friends are central, as responders during pandemics are being exposed to infected patients at work [10, 13, 21–23]. In a study conducted by Goulia et al. [13] on the H1N1 influenza outbreak, the most frequent concern of health care workers was the possibility of infecting family and friends and the health consequences of the disease. The same concern among health care workers is prevailing during the current COVID-19 pandemic and might be especially true for staff living with people considered to belong to a risk group. Cai and colleagues [21] show that staff between 31–40 years of age had the greatest concern regarding viral transmission to their families. A possible explanation is that more people in this age group are living together with older people and children. Also Dong et al. [17] found that healthcare workers who had fears about their own physical health and who had friends or relatives tested positive for COVID-19, had a higher probability of anxiety and depression. Concerns of spreading the virus are also fuelled where there is limited access to testing and/or personal protective equipment (PPE), which makes the own risk assessment of health care personnel somewhat more difficult [24]. Not being rapidly tested when developing symptoms may not only lead to fear of spreading the disease among friends and relatives but also among patients and colleagues at work [24].

## Decreased feeling of safety and loss of trust

Furthermore, a decreased feeling of safety due to limited access to appropriate personal protective equipment seems prevailing [3, 22]. A feeling of unsafety is also grounded in often not having access to up-to-date information and if clear guidance is not communicated

accordingly. During the H1N1 outbreak, personnel with better access to information scored lower on stress symptoms than those with limited access [14].

## Social stressors

Social factors inherent in the everyday private life of health care workers cause difficulties during pandemics. Not being able to access childcare or to provide support for other personal and family needs might be a barrier to working [25]. Furthermore, in many cases health care workers feel stigmatized, as they are avoided or treated differently due to high exposure to infected patients [12, 13, 26].

## Quarantine experience

Studies suggest that being quarantined may have a more severe impact on the group of health care workers than other population groups. However, the authors argue that this effect may not be unique to their health care workers status but may be influenced by their job-related experiences, which results in recognition of higher personal risk or more knowledge regarding the severe end of SARS on the clinical spectrum. In their study, health care workers also had a better understanding for the rationale for quarantine and showed more compliance with quarantine behaviors [27]. Several studies on health care workers affected by the SARS outbreaks in Canada, China and Taiwan found quarantining experience to be predictive of subsequent general psychological distress [9, 11, 28, 29]. However, in a study by Hawryluck et al. [30] on the psychological effects of quarantine in Toronto during the SARS outbreak, health care worker status was not correlated with negative effects such as PTSD or depressive symptoms.

## Moral injury and ethical dilemma

Several researchers state that moral injury is one of the mental health challenges faced by health care workers during the COVID-19 outbreak [31–33]. According to Shay [34] the term moral injury "has been used in two related, but distinct, senses; differing mainly in the 'who' of moral agency". According to Shays own definition, moral injury is present when there has been (a) a betrayal of 'what's right'; (b) by a person in legitimate authority (e.g. a leader) and, (c) in a high stakes situation. The definition of Litz et al. [35] also entails participation by one's self in acts that transgress such moral beliefs.

Both forms affect trust and lead to psychological distress. Moral injury has mainly been described in victims and perpetrators of violence such as child soldiers. However, not only intentional interpersonal disasters may involve moral injury, also other disasters like pandemics include these types of trauma. ln the COVID-19 pandemic, health personnel faced a variety of those injuries in both of the above mentioned meanings.

In a global pandemic, a different set of rules has to be applied to healthcare delivery as complex dilemmas in care may evolve. Understanding general principles of collective ethics may help, nevertheless clinicians still have to take decisions for their specific patients, which can lead to significant distress [31]. The ethical bitterness of triage decisions is well-known and medical organisations and professionals have found different ways to deal with it [36, 37]. Some of the most common ethical challenges in the response to COVID-19 can be conceptualized in triage, shortage of personal protective equipment and non-pharmaceutical interventions. The following strategies can be identified to deal with the dilemmata [38–40].

**Triage.** While cost benefit analysis in the utilitarian model of resource distribution is based on criteria such as numbers of lives saved, number of years of life saved or quality of life that might not be compatible with human rights regulations, other criteria have to be found to allocate resources such as the criteria of medical need and efficacy of treatment.

**PPE shortage.** Regarding the protection of EMS clinicians [41] and of health care workers, a balance between duty to care and protection of health care workers has to be found. Especially in a situation in which the availability of health care workers is crucial, they should have priority in the distribution of resources.

**Non-pharmaceutical interventions.** All non-pharmaceutical interventions that are critical (with regards to e.g. data protection), need to be appropriate with regard to necessity, proportionality and minimization (of e.g. data use).

## Effective support

Several factors that are specific to the pandemic context have been found to help in reducing stress among health care workers. Strengthening individual as well as organizational resilience can help in mitigating the negative long-term influence of stressful experiences among health care professionals [26].

## Altruism

While being at risk, studies show that moral and social responsibilities as well as altruistic attitudes drive health care workers to continue working in an environment that might be extremely stressful [9, 21, 23, 25]. Data showed, that altruistic risk-acceptance during the SARS outbreak could decrease the odds of higher levels of depression-symptoms three years after the outbreak [9]. Wu et al. also identified altruism as a protective factor against negative impacts. Underlining the valuable altruistic attitude can reduce psychological distress especially in those who are quarantined [10]. Altruism being a protective factor, elements that reduce altruism and endanger one´s self view as a helper may do a lot of damage for health care personnel.

## Guidance and shared decision making in ethical dilemma situations

Receiving good guidance and not having to take decisions on their own as well as good leadership may minimize the impact of moral injury. Shared decision making and tools may thus be very helpful to mitigate negative impact [31, 32].

## Information

Information and efficient and fast communication are essential elements in order to decrease worry and promote a feeling of safety among staff [13]. This includes information about the virus, routes of infection, possibilities and practical guidance for treatment or protection measures, but also factual information on the situation within the given hospital, such as capacities or number of infected people among staff [18].

## Training

Another stress-reducing factor is training and support in using PPE. The provision of equipment, including masks and suits as well as infection control guidance by the hospital leadership have been reported to promote a feeling of safety in the MERS-Cov outbreak [23] as well as SARS [29] and other outbreaks. Marjanovic et al. [11] showed that trust in infection control initiatives as well as protective equipment predicted lower levels in correlates of burnout and stress among nurses engaged in the SARS outbreak in Canada. Raising awareness of the effects of disease prevention measures among staff with reduced numbers of reported cases can reduce staff stress [21].

## Psychosocial interventions

Furthermore, psychosocial interventions may be more helpful than support that is too much focused on clinical psychological interventions [22]. Psychosocial services among health care workers provided either by peers or by well-known and trusted mental health professionals—such as psychologists, clergy or psychiatrists—have been shown to be especially helpful [10]. Signalling in-group messages that all staff are in this together and nobody has to carry decisions alone are considered as very stress relieving [9, 10, 21].

Opportunities for psychoeducation and psychosocial counselling are essential for the protection of personnel. However, during the pandemic the provision of psychosocial support has been reported to be most effective when including hospital management and leadership and focusing on pragmatic support based on the actual needs of the staff. This can be done by taking care of basic needs, such as break and resting areas, food, daily living supplies and ways to be in contact with their families [22].

Pre-job-trainings on identification and responding to psychological problems in oneself and others, as well as on coping with stress have been reported as helpful during the SARS outbreak as well as the COVID-19 situation in China [22, 42]. The acceptance and use of psychological support might be higher when provided low-threshold access. Counsellors visiting break and resting areas where stories and difficulties are shared, and providing support accordingly is one way of providing low-threshold access [22].

In many cases, anonymous helplines and counselling have been established in order to reach highly affected staff in the hospitals. However, considering the access to psychological services, it should be taken into account, that staff with subthreshold and mild levels of mental health disturbances might actually be more likely to take action and be motivated to acquire skills to help others, than those with more severe disturbances who have more motivation to learn self-help techniques [3].

## Resilience

If we look at the concept of resilience in the context of natural hazards we can identify three core elements: resistance, recovery and adaptive capacity [41, 43]. Resistance is the ability of a system to withstand a threat, which is mostly a consequence of strength and preparedness. Recovery is the ability to bounce back and be able to come back to a state of normal functioning after a disaster has hit and adaptive capacity refers to the ability to learn and change due to the experience made [41].

After the SARS outbreak in Canada, Maunder et al. [42] did a study on resilience in healthcare workers. Although the authors recommend special trainings for healthcare workers, evidence points in the way that resilience building on the organisational level is of utmost relevance for pandemics. This goes way beyond the development of adequate training and psychosocial interventions, although these are also very important to have.

According to the authors, two constructs have been relevant for organizational resilience in a pandemic: Organizational justice and "magnet hospitals". Organisational justice refers to the amount to which leadership and management take the opinions of healthcare staff into account and take their concerns seriously. The first element of organizational justice is relational justice, the ability of team leaders and managers to suppress their own prejudice and treat their staff in an honest and just manner. The second element is decisional justice, which mainly refers to fair decision making. Organisational aims, which are based on the needs of patients as well as the needs of healthcare workers, lead to more (crisis) resources.

Another relevant construct for organisational resilience was found in so called "magnet hospitals". These were characterized by decentralized decision-making and nurses being

amongst administrative staff and management, a flexible approach to shifts and continuous effort in training and self-administration of units.

These results confirm other findings from organizational psychology that have shown that a high amount of demands mixed with low control (low decision making capacity and low influence in staff) and an imbalance between effort and reward has negative effects on staff health and wellbeing [44, 45].

## Methods

In this study, we are trying to identify the subjective experience, stressors and collective coping strategies used in hospital care to relieve stress from a psychosocial perspective and manage the challenging situation of the pandemic. The objective of the study was to find out what the main stressors as well as the main stress reducing factors were during the first phase of this pandemic for healthcare workers in the hospital sector. As many of the psychosocial challenges during the pandemic have to be considered from an ethical perspective, we were also interested in ethical questions in COVID-19. Amongst others, these included resource distribution when there is shortage in supplies as well as the duty to help while risking one's own health and health of relatives considered vulnerable. We assumed to find indicators of moral injury in healthcare workers to be of importance during this first phase of the pandemic. However, we were not only interested in the negative effects of the pandemic on healthcare workers. We were especially interested in (organizational and team) resilience, namely indicators for resistance, recovery and adaptive capacity of healthcare workers themselves as well as the systems and organisations they work in.

The data presented in this article is part of a larger study done in the course of an EU Project (NO-FEAR) where we use a mixed methods approach alternating between quantitative (online) surveys and qualitative interviews and focus groups. The institutional ethics committee of the Institute for Psychology at the University of Innsbruck as well as the NO-FEAR Project External Ethics Advisory Board reviewed and approved this study.

As a first step, we chose to conduct expert interviews. The reason for this choice was the fact that we expected an explorative approach to be the best way to gain more detailed insights in both, the positive and negative subjective experiences of healthcare workers during this first and very stressful phase of the disaster. As we did our study very early in the crisis we knew that healthcare workers were still in the middle of a very stressful job and expected them not to be willing to do long interviews or fill out questionnaires. We thus focused on a small number of experts working in the field including mental health professionals as well as experts from the medical area. According to the principle of Grounded Theory [46], we aimed at maximum contrast and thus tried to get interviews from rather different contexts and backgrounds. Our main factors for the choice of experts was that they were very experienced, working in the field and in a leading position or a position that allowed to speak for their colleague healthcare workers from an expert position.

Our aim was to develop some basic hypotheses regarding resilience and stress experience of healthcare workers in the first phase of the COVID-19 pandemic. These hypotheses shall be a basis for the interpretation of data from questionnaires and interviews gained in later phases of the pandemic. We are well aware that these data are far from generalizable, but expect our hypotheses to be a good basis for further research as well as practical conclusions.

In total, 13 experts were interviewed by two teams. Participation of experts was requested via telephone calls. Eligible experts included in the study sample had to be health care workers involved in the current COVID-19 response either in hospital settings. Health care staff had to have more than 5 years of experience. Psychologists and intensive care physicians had to have

more than 10 years of experience. This included doctors, nurses and psychologists in leading positions in Italian and Austrian hospitals. All professionals interviewed were informed about the scope of the interview and their rights. They subsequently signed an informed consent that was previously reviewed by the NO-FEAR Project External Ethics Advisory Board.

The Austrian team consisting of male and female scientifically experienced psychologists conducted two semi-structured interviews with mental health experts from hospitals in Italy and one Focus group discussion with four psychologists involved in the hospital response in Austria. The Italian team, consisting of female scientifically experienced researchers in the medical field conducted 7 semi structured interviews with healthcare professionals from the hospital, revising the questions based on the experience and the role of each interviewee. Questions focused on the main challenges that health care workers experienced in the acute phase of the COVID-19 response as well as the needs they have. Additionally, we focused on strategies that health care workers experienced as helpful or stress relieving to identify possibilities for psychosocial support measures.

One additional interview was conducted on challenges from the ethical and legal perspective in hospital care with Laura Palazzani [40].

Data was collected from the beginning of March, when the virus began to rapidly spread in Italy and Europe until the end of May.

Table 1 is giving an overview on basic demographic data of the study sample.

The interviews were transcribed and eventually analysed using Grounded Theory methodology. Data was discussed with participants to allow for comments and correction. We coded data into several concepts and subcategories to identify psychosocial and ethical challenges during the COVID-19 response. Codes were subsequently related to each other using Axial Coding. We created a basic explanatory framework on psychosocial effects of COVID-19 consisting of subjective stress experience, causal factors, stressors and resilience factors during the response.

## Results

### Psychosocial effects of COVID-19

The following paradigmatic model shall give an overview of the main categories and their links. Causes, intervening stressful and stress reducing factors and coping strategies are

**Table 1. Study sample including information on function, gender and country of origin of interviewees as well as the method used to gather information.**

| Occupation/function | Gender | Country | Method |
|---|---|---|---|
| ED intensive care physician | m | Italy | interview |
| Head nurse of ED | f | Italy | interview |
| Psychologist | f | Italy | interview |
| Psychologist | f | Italy | interview |
| Psychologist, Head of unit | f | Italy | interview |
| Director of intensive care unit | m | Italy | interview |
| Trainer for HCW protection and PPE, chief registered nurse | f | Italy | interview |
| Psychologist | f | Italy | interview |
| Psychologist | m | Italy | interview |
| Four Psychologists | m/f | Austria | Focus group |

m = male; f = female; ED = Emergency Department.

gathered around the phenomenon of interest namely the subjective stress experience of health-care personnel.

## Key category 1: Fear, guilt feelings, frustration, loss of trust and exhaustion

Emotionally, fear of getting infected and subsequently infecting families and friends are common feelings. Experts point out the prevalence of feelings of guilt and shame as employees feel like being "plague spreaders".

> "...staff is scared because they live with kids or elderly/immunodepressed people. They are concerned to take the COVID home. Normally they avoid to bring home problems or issues related to their job but now they feel they can bring home COVID but also their feelings. ED staff also try to avoid to send mothers or fathers of kids to the infected patients"(Interview partner 1, Italy).

This resulted in many health care workers not returning home as they did not want to put their own families at risk. Some staff have been reported to be sleeping with masks at home. Many health care workers feel unsafe and experience a loss of control, as well as a loss of trust in the system, as they do not feel sufficiently protected.

Frustration and powerlessness are prevailing. Among nurses, rage and demands for appropriate compensation is central. Among doctors, more silent reactions seem common which has been described by one interview partner as a "*paralysis of thought*".

> "*Nursing staff is reacting with rage to the stress the COVID 19 is generating, while doctors are reacting with silence, stunned silence (it has been defined by some as a "paralysis of thought")*" (Interview partner 3, Italy)

One other interview partner points out *"that fear of infection among health care workers is higher after the shifts when resting at home" (Interview partner 1, Italy).*

Socially, stress reactions included increased lack of trust in external or newly hired workers, as more experienced staff are afraid that unexperienced workers might make mistakes. But also a loss of trust in the hospital and in oneself as well as a need for a caring leadership was observed.

> "*Hospital staff is suffering from a sense of abandonment from the institution (which includes medical direction of the Hospital and in some cases also by the chief physician as well as the chief nurse). They are looking for someone (above mentioned) that nurture them and would like to have a deeper recognition of the work they are performing and more gratifications. One aspect that is frequently mentioned is the "sharing of moments", be they for discussion of clinical cases to decision-making and especially directives on how to relate to the patient and the relatives. The relationship the personnel is looking for is more similar to a child-parent scheme than subordinate-superior scheme.*"(Interview partner 3, Italy).

Physically, exhaustion and fatigue are prevailing. This is also true for staff in the laboratories due to high amounts of positive blood test results. Psychologists report about insomnia as a common issue of health care workers according to the experts that were interviewed.

Cognitive stress reactions include confusion and unrest as well as high levels of dissociation among health care workers involved in the COVID-19 response. One expert points out signs of dissociation among health care workers, stating "*It's like a film scene from a war, just without visible destruction*" (Interview partner 5, Italy).

Also positive emotions were reported. Pride of having managed a difficult situation together as well as a feeling of high team cohesion and solidarity among the healthcare personnel and a deep commitment to their job as a healthcare worker have been mentioned by many experts. This also led to a feeling of "us" versus "them" meaning "us frontline workers" versus "them in the background".

## Key category 2: Causal factors: Rapidly evolving situation with high uncertainty

The COVID-19 pandemic is a rapidly evolving situation in which the protection of the population, but especially of people with high exposure to positively tested patients, such as health care workers in the hospitals, is central. The situation in the first wave of this pandemic was characterized by the uncertainty of information and lack of knowledge about the virus and its characteristics. Additionally, the rapidity of change in information as well as the rapidity with which patients tested positive came into hospitals and/or the rapidity of change in their medical conditions is pivotal. A further characteristic is the fact that one may infect others also when not (yet) having symptoms. This increases insecurity. Factual or anticipated lack of resources such as unsafe or damaged PPE, as well as lack of knowledge when the first wave or pandemic as a whole will end, increased stress for healthcare workers.

*"One main stressor was the communication and information of colleagues, the permanent and fast changes were very challenging, often the overview was lost what is or is not the present procedure lead to different ways of action in some cases. . ." (Focus group member 4, Italy)*

## Key category 3: Stressors

**Stressor 3.1: New roles/tasks and broken routines.** Healthcare workers often had to work in newly formed teams that consisted of staff that did not know each other well. A feeling of powerlessness was often grounded in the fact that new roles had to be carried out, many of which health care workers did not feel appropriately prepared for. Broken routines and lack of opportunities for social exchange were another common stressor. Normal social exchange during breaks as well as common routines could not be used during the shifts. Stress is also grounded in the unavailability of "safe places". In many cases there was no space for social support or to share experiences such as in common rest areas. One expert pointed out that

*". . .all the small relieving things that staff are used to were missing, such as sitting and eating together".*

**Stressor 3.2. Working with PPE.** Usage of PPE in itself is a stressor. High temperature, uncomfortable usage, resulting skin and other health problems are some of the distressing factors. Additionally, health care workers did not drink during work and did not use the toilet often for 12 hours and more because PPE has to be exchanged when taking a break. Many workers were working voluntarily overtime, leading to less time for adaptive processing of emotions.

**Stressor 3.3. Loss of professional distance, changed relationship with patients and relatives, moral injury.** Moral injury has been expressed by almost all experts. As expected, Triage and PPE shortage were the most prominent stressors. In many cases, staff complained about shortage in supplies, PPE, medications or replacement parts of instruments. This increased the feeling of insecurity among health care personnel. In some cases, scared staff

hoarded material for personal protection, which aggravated in further imbalance between demand and supply of equipment. In other cases, lack of resources has not been a major issue. However, even then the anticipated lack of resources created a lot of anxiety and stress. Triage decisions, as well as decisions regarding the distribution of PPE were experienced as extremely stressful especially by the medical doctors involved.

Further indicators of moral injury were mentioned such as lack of being protected, missing or damaged PPE, authorities and leaders knowingly endangering staff, fear of staff to infect others, dealing with dead bodies without the religious rituals normally used or isolation of dying patients from their relatives. All these stressors together created a climate where trust in the system as well as the self-image as a good person were endangered.

The situation was enhanced by a loss of professional distance: Many workers were mediating between families and patients, as relatives were not allowed to visit. These tasks differed significantly from the daily routines that staff were used to and forced the healthcare workers into an unusual intimacy towards patients and relatives that endangered their professional distance. Many healthcare workers were providing psychosocial support to patients who were dying alone in the hospitals. In these cases, part of their professional distance got lost, as they were not able to use the protective strategies they normally use. Due to protective measures, the relationship to patients was experienced differently. Healthcare workers experienced PPE as limiting communication and trust building especially in relationships with "strangers".

> "….*patients are dying alone. Few members of the staff enter in the patient room and say "this patient is going to die, what should I do for him/her*?*" Clearly this is totally different from what they learnt and what is normally happening. They feel like they have the role of mediation and communication between the patient and the family (staff help patients in talking on the phone with relatives)" (Interview partner 1, Italy)*

> *Nurses had to care for dying patients without their routines (having relatives and friends present). Often they were the last ones to see the dying patient and relatives often want to talk to them as they have been the last ones who have talked to their dead relative" (Interview partner, 10, Italy)*

Many health care workers experienced stigma as they were avoided or treated differently because of their high exposure to COVID-19 patients. They were perceived as dangerous in their private environment, whilst at the same time being treated as heroes in their professional environment and in the public opinion.

> *"they felt like virus spreaders"(Interview partner 3, Italy)*

For many health care workers appropriate risk assessment was difficult, as staff was not tested regularly in the beginning because of test shortage. Many workers were worried of infecting relatives they considered as vulnerable such as children and older relatives living with them at home. As a reaction, many healthcare workers did not go home to sleep and avoided their families. This strategy of prevention robbed them of the most important resource for recovery.

## Key category 4: Resilience factors

According to the experts, health care workers have proved to be a very resilient group. Despite major stressors such as being in new teams and having to deal with dying patients and their families as well as the many experienced threats and moral dilemmas, healthcare personnel

adjusted to the new situation and roles and found new routines. Many expressed their pride to have managed a difficult situation together as a team. Regarding social exchange, healthcare workers found new ways to interact with and support each other during the beginning or end of the shifts or during training sessions as well as during shifts when working in pairs.

**4.1 Organisational resilience: Leadership, dialogue, protection and guidance.**   Nevertheless, "external" support by leadership, peers and mental health professionals have proven once more to be of utmost importance. From an organizational point of view, a strong and trusted operational leadership, backed up by multi-professional crisis management staff in the background providing support and strategic guidance, was experienced as the most valuable support.

Leadership takes a central role in how challenges are experienced by health personnel. Several experts pointed out, that clear guidance provided by the management led to more stable routines and therefore was experienced as stress relieving. There was a need for regularly updated guidelines for protection of health care workers as well as accurate information and training e.g. on the use of protective equipment. The better the communication worked and the more the messages were experienced as clear, the better they were received by the personnel. One expert pointed out, that

"*written information wasn't read and thus did not reach all health care workers, while recorded videos by the management were much more efficient with hundreds of employees watching (Interview partner 5, Austria).*

Another strategy that was well received was holding regular meetings of team leaders on the balcony in order to exchange important information. Pre job trainings for everybody held face to face in small groups and visits by a mixed team of hygiene experts and mental health professionals directly at the units before shifts in order to support them with PPE use were experienced as very helpful.

Enabling a dialogue between management and staff was very much appreciated. In some hospitals, feedback mechanisms were established. For example, one person from the management was made responsible for "listening to staff" and to take care that reported needs and problems were addressed in a timely manner. These were often just "little things", but of a high importance to health care worker´s wellbeing. Possibilities to ask questions online were established, so that leadership could react e.g. with information provision via videos. Having hygiene teams sometimes mixed with psychologists available in the wards at the beginning of shifts also seemed to be helpful to reassure personnel in the usage of PPE and provide a sense of safety and appreciation.

Protection of health care workers including sufficient materials, as well as clear guidance and training on how to use equipment has been reported as one of the most important strategies in order to relieve stress for health care workers. In some hospitals, workers with children at home were not sent to work in Emergency Departments. When possible, working in pairs promoted reassurance and thus reduced feelings of insecurity in difficult tasks also because there is mutual help in donning and doffing PPE. Clear directions on how to relate to severely ill patients and relatives were experienced as helpful. Additionally, possibilities to exchange among peers on clinical cases and decision making as well as sharing of special moments has been pointed out as helpful to promote peer support and relief stress among staff. This often asked for an adaptation of areas where people can meet and exchange while keeping the necessary distance, like outside areas or larger rooms.

Guidance and joint decision making regarding triage has reduced a lot of stress. One example is a pro forma "validation" of each patient before a shortage appeared in a medical team

according to criteria that have been developed by a mixed team of medical, ethical and psychosocial experts and approved by the management. Thus, in situations when a rapid decision had to be made, for example during a nightshift, the person taking the triage decision had a basis from which to start and did not feel completely left alone.

**4.2 Mental health and low threshold psychosocial support.** Mental health support was described as most successful when medical leadership and mental health leadership were working openly together. However, mental health support was directly requested from the healthcare workers mainly for patients and relatives.

In many cases, Psychosocial Support did not only include psychologists but also hospital clergy was directly requested when help was needed in mediation between patients and families or in supporting severely ill or dying patients. Direct face to face support as well as coaching of healthcare staff how to deal with families and patients given by clergy, psychiatrists or psychologists, was very well received and reduced stress in healthcare staff. Group interventions for staff were often not possible as staff was too overworked to attend. However, free and anonymous service for staff is essential in providing low threshold support for all staff in the hospitals and also in pre-hospital settings in which anonymous helplines have been well received. Proactive contact to healthcare personnel who had to stay at home in quarantine has also been reported as well received but only when done by someone known and trusted.

Focus on the actual and practical needs of health care workers in the acute phase has been reported to be interventions that are more successful than classical clinical approaches. For example meeting the need for social gatherings (e.g. meeting once a week on the balcony or during trainings in a big room) as well as support in providing small islands of normalcy and common rituals (e.g. Easter) were experienced as extremely helpful. Another example was setting up a piano in the ward for certain hours in order to confront silence.

Psychologists and/or clergy visiting the wards in an outreaching manner was described as much more effective as remote forms of support. Direct contact by mental health professionals and management has been highly appreciated by staff. One expert even spoke of a "knight blow" for the psychosocial staff and several experts pointed out that medical staff did not take it well if mental health professionals stayed in the background only.

## Discussion

The output of this article is directly related to the first one and a half months of the COVID-19 pandemic and healthcare crisis in Italy and Austria. The evolving situation will probably lead to different conclusions that will be drawn during the "lesson learning process", that will take place at different levels in all the healthcare domain.

However, this output has to be viewed as a gathering of first level immediate impressions and experiences from healthcare staff involved in this massive healthcare crisis. Like a ship's logbook it contains the day to day evolution and perception of the healthcare personnel through the lens of mental health and medical experts who themselves were deeply involved into the mission of "fighting corona".

In knowledge management, lesson learning is a process that is usually done after the crisis with a debriefing mechanism and having players of multiple roles sitting at the table to draw common conclusions. Lesson learning is usually defective in the part that deals with retrieving and analysing first-hand experiences. We do not want this to happen for the COVID-19 crisis for what pertains to the in-hospital personnel psychosocial response.

The importance of this work can be summarized in the following three points:

1. Gathering of first-line experiences and impressions

2. Steering the steps of a future lesson learning process

3. Precociously identifying issues that due to the long timeline of the COVID-19 crisis might be of help to countries whose response will be postponed in time as well as to all those who are faced with further waves of the pandemic.

One of the main hypotheses that we derived from our data is that moral injury seems to be a common and central outcome of this crisis in many healthcare workers, be it in areas where the virus has had an extreme negative impact on the healthcare system or in areas where the pandemic had not so much negative impact (yet). Even though extreme measures like triage did not have to be applied in all areas, other challenges to healthcare workers' ethical self-image and worldview have been badly shaken by this crisis. To be forced to separate dying patients and their families, to have to handle a dead body without the usual rituals, to be not only a helper but at the same time a possible threat to your own colleagues and family have been experiences shared by many healthcare workers during this pandemic.

Our second main hypothesis is based on the findings that organizational justice as well as a flexible decentralized approach to leadership lead to more collective and organizational resilience during this pandemic. Our findings support the assumptions of Greenberg et al. [32] or Williams et al. [33]. Loss of trust in oneself as a helper as well as loss of trust in the system or in the leadership are common effects of moral injury. From a practical point of view, the challenge that we are facing now after the first phase of the pandemic is to rebuild trust in ourselves as helpers as well as in the systems that we are working in. As Maunder et al. have stated in 2008, those hospitals that have a structure of flexibility, decentralized decision making as well as continuous training and self-administration of units fared better during this crisis as highly centralized organisations with an administration that did not understand the needs and concerns of the staff "on the ground" [42]. Furthermore, as Lancee et al. [26] also stated, leadership and organisational justice played an important role in counter balancing moral injury. Being taken seriously in their concerns as well as visible efforts to take just and well-grounded decisions were main resilience factors also in our experts´ view.

Our third hypothesis is that psychosocial support of healthcare workers during this pandemic only works if it is given by trusted mental health professionals or peers in a very basic manner integrated into the overall support by the management (trainings, information, frameworks provided). The best support given during the first phase of a disaster response is on scene support guided by the motto 'to act with the people and not for the people' [47]. Thus, mental health professionals who acted with the healthcare personnel and not for them (from an external safe position) and who acknowledged and appreciated the resilience that this group already has and did not treat them as potential patients were the only ones whose support was accepted.

Williams et al. [33] recommend several strategies to deal with moral distress and moral injury faced during COVID-19. They base their recommendations on the well-known elements of effective psychosocial support: safety, connectedness, self and collective efficacy, calm and hope [48]. Authors such as Juen and Warger [49] as well as Dückers et al. [50] totally agree with this. Hobfoll et al. [48] mention the need to translate the elements into the given context of each disaster.

## Conclusions

According to our data, safety refers to good, honest and timely information given most directly by trusted leaders. Sufficient PPE but also visible efforts to support and protect staff by the management as well as interest in how they manage the situation leads to a feeling of safety in

a situation where many healthcare workers had the feeling that there is no safe place left. Paradoxically, those who worked in the COVID units often mentioned to feel safer than those in other units where no one knew when a patient could be tested positive. This was especially true during the very first phases when PPE was not available for everybody. Connectedness refers to efforts of staff themselves and their leadership as well as mental health professionals and peers to enhance group cohesion and allow for social exchange in spite of difficult circumstances. Examples were meetings on the balcony after the shift for a quick exchange. Connectedness also was expressed by proactive contact to those who had to stay at home because of infection or quarantine. Calm was reached by providing enough space for rest and recovery (e.g. accommodation when staff does not want to go home). Calm was also reached by re-establishing normalcy including common rituals during holidays or other efforts to establish a normal working environment. Self and collective efficacy has been reached by preparing staff for their new tasks, taking joint decisions for example by discussing ethical guidelines together and trying to find joint solutions. Caring for dying patients and connecting with their families have been main stressors, but also sources of self-efficacy and pride. The element of hope refers to the fact, that healthcare staff, despite their distress, exhaustion and many sacrifices, went through this crisis together and managed the task successfully. From what we have learned, we can conclude that many of the healthcare workers have shown an extraordinary resistance. They have been able to work together and function well as teams in very challenging circumstances (new tasks, new teams, new challenges, rapidly changing environment). They have often overworked themselves until they were completely exhausted. Many of them took damage to their health or lost their lives due to the pandemic. Faced with extremely distressing ethical dilemmas, they have been able to take decisions based upon rational models and guidelines while at the same time providing the best care possible for their patients in these difficult times.

## Supporting information

**S1 Checklist. COREQ checklist.**
(PDF)

**S1 File. Psychosocial considerations for health care workers in the COVID-19 pandemic.**
(DOCX)

**S2 File. Ethics statement by the institutional review board of the University of Innsbruck and external ethics review board of the NO-FEAR project.**
(PDF)

## Author Contributions

**Conceptualization:** Alexander Kreh, Rachele Brancaleoni, Sabina Chiara Magalini, Barbara Juen.

**Data curation:** Alexander Kreh, Barbara Juen.

**Formal analysis:** Alexander Kreh, Rachele Brancaleoni, Sabina Chiara Magalini, Barbara Juen.

**Investigation:** Alexander Kreh, Rachele Brancaleoni, Sabina Chiara Magalini, Barbara Juen.

**Methodology:** Alexander Kreh, Barbara Juen.

**Supervision:** Barbara Juen.

**Writing – original draft:** Alexander Kreh, Rachele Brancaleoni, Sabina Chiara Magalini, Nils Ellebrecht, Barbara Juen.

**Writing – review & editing:** Alexander Kreh, Rachele Brancaleoni, Sabina Chiara Magalini, Daniela Pia Rosaria Chieffo, Barbara Flad, Nils Ellebrecht, Barbara Juen.

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
