## [Decision Letter · Decision Letter 0]

12 Jan 2021

PONE-D-20-34707

Ethical and Psychosocial considerations for hospital personnel in the COVID-19 crisis: Moral Injury and Resilience

PLOS ONE

Dear Dr. Kreh,

Thank you for submitting your manuscript to PLOS ONE. After careful consideration, we feel that it has merit but does not fully meet PLOS ONE’s publication criteria as it currently stands. Therefore, we invite you to submit a revised version of the manuscript that addresses the points raised during the review process.

We look forward to receiving your revised manuscript.

Kind regards,

Francesco Di Gennaro

Academic Editor

PLOS ONE

Journal Requirements:

2. Please include additional information regarding the interview guide used in the study and ensure that you have provided sufficient details that others could replicate the analyses.

For instance, if you developed an interview guide as part of this study and it is not under a copyright more restrictive than CC-BY, please include a copy, in both the original language and English, as Supporting Information.

In addition, we recommend that authors use the COREQ checklist, or other relevant checklists listed by the Equator Network, such as the SRQR, to ensure complete reporting (http://journals.plos.org/plosone/s/submission-guidelines#loc-qualitative-research). Please attach such a checklist as Supporting Information.

Additional Editor Comments:

dear authors follow reviewer suggestion to improve your paper

Reviewers' comments:

Reviewer's Responses to Questions

**Comments to the Author**

1. Is the manuscript technically sound, and do the data support the conclusions?

Reviewer #1: Yes

Reviewer #2: Partly

2. Has the statistical analysis been performed appropriately and rigorously? 

Reviewer #1: N/A

Reviewer #2: N/A

3. Have the authors made all data underlying the findings in their manuscript fully available?

Reviewer #1: Yes

Reviewer #2: No

4. Is the manuscript presented in an intelligible fashion and written in standard English?

Reviewer #1: Yes

Reviewer #2: Yes

5. Review Comments to the Author

Reviewer #1: the study was a descriptive based on intensive interviews and was writing in clear standard English understandable for scientific community as well as lay person everything was technically sound and I have no comments on it

Reviewer #2: 1.Only 13 experts were interviewed in this study. Is it fully representative and statistically significant？Whether there is a bias in the selection of study object？

2.Please provide the interview outline and questionnaire.

6. PLOS authors have the option to publish the peer review history of their article (what does this mean?). If published, this will include your full peer review and any attached files.

Reviewer #1: No

Reviewer #2: **Yes: **Dengchuan Wang

---

## [Author Response · Author response to Decision Letter 0]

12 Mar 2021

Dear Editors, 

dear Reviewers,

Thank you for reviewing our submission entitled Ethical and Psychosocial considerations for hospital personnel in the COVID-19 crisis: Moral Injury and Resilience (PONE-D-20-34707).

Attached to this letter we are uploading a marked-up copy of our manuscript that highlights all changes made as well as an unmarked version of the revised paper.

In the following, we want to respond to all points raised by the reviewers.

1. Style requirements

We updated the manuscript according to PLOS ONE’s style requirements. 

2. Additional information

We now include our interview guide in the supporting documents section.

Given that we conducted our research in the very beginning of the COVID-19 pandemic, an exploratory approach was necessary to gain insights on the health care worker’s perspective in this new environment. We thus followed the principles of openness and induction, as we stated in the methods section of the revised manuscript. Our interview guide contains topics that provide a framework of orientation to ensure comparability and accompany the communication process. However, instead of asking specific, predetermined types of questions the guide relies on the interaction with the interviewee to steer the interview process and helps constructing questions on the topics of interest as the interview progresses – a method that is widely used in qualitative research (see e.g. Witzel, 2000; Edwards & Holland, 2013, p. 55)

We also now provide the COREQ checklist to ensure that we addressed all points relevant for complete reporting. 

3. Data availability statement

We were asked to make all data underlying the findings in our manuscript fully available.

However, in our first submission we pointed out concerns as our interview transcripts contain sensible participant information. Publishing the full transcripts is also excluded in the informed consent our interviewees signed. In order to meet transparency needs, excerpts of the transcripts relevant to the study are made available within the paper.

We also provided an overview of our conceptualized data as in this form it does not contain sensible participant information. It is available in a public repository:

“Full data cannot be shared publicly because they contain potentially identifying or sensible participant information. However, conceptualized data after using grounded theory methodology can be retrieved from doi.org/10.5281/zenodo.4241858”

4. Additional review comments

Reviewer #2 pointed out concerns regarding our sample size, whether there is a bias in the selection process and whether it is fully representative and statistically significant. 

We would like to point out the inherent principles of sampling approaches in qualitative research. We did not use a random sampling approach in order to find a statistically representative sample (as it is indispensable in quantitative research). Instead, we purposefully selected information-rich cases, aiming at in-depth understanding rather than statistically proven generalizations (see e.g. Patton, 2002, p.230). We thus do not view our sample as biased, but consider the selection of highly experienced interviewees directly involved in the response of the pandemic as a quality criterion of our data. 

We furthermore believe that our approach is in line with trends of narrative-based medicine in recent years (see e.g. Kalitzkus & Matthiessen, 2009).

Our aim in this manuscript was to develop some basic hypotheses regarding resilience and stress experience of healthcare workers in the first phase of the COVID-19 pandemic. However, we agree that a statistical survey would add to the value of our findings. We are therefore currently conducting survey studies with health care workers. The construction of the surveys is based on the findings from qualitative research presented in this manuscript. Findings from quantitative follow-up studies will be presented at a later stage. 

Sources

Edwards, R. & Holland, J. (2013). What is qualitative interviewing? Bloomsbury Academic.

Kalitzkus, V. & Matthiessen, P. (2009). Narrative-Based Medicine: Potential, Pitfalls, and Practice. The Permanente Journal, 13(1), 80-86.

Patton, M. Q. (2002). Qualitative research & evaluation methods. 3rd (ed.). Thousand Oaks, CA: Sage.

Witzel, A. (2000). The Problem-Centered Interview. Forum: Qualitative Social Research, 1(1), Art. 22. Available at https://www.qualitative-research.net/index.php/fqs/article/view/1132/2521

Thank you for taking our manuscript under further consideration.

Best regards,

Alexander Kreh 

Corresponding author

---

## [Decision Letter · Decision Letter 1]

22 Mar 2021

Ethical and psychosocial considerations for hospital personnel in the COVID-19 crisis: Moral injury and resilience

PONE-D-20-34707R1

Dear Dr. Kreh,

We’re pleased to inform you that your manuscript has been judged scientifically suitable for publication and will be formally accepted for publication once it meets all outstanding technical requirements.

Kind regards,

Francesco Di Gennaro

Academic Editor

PLOS ONE

Additional Editor Comments (optional):

dear authors congratulations

Reviewers' comments:

Reviewer's Responses to Questions

**Comments to the Author**

1. If the authors have adequately addressed your comments raised in a previous round of review and you feel that this manuscript is now acceptable for publication, you may indicate that here to bypass the “Comments to the Author” section, enter your conflict of interest statement in the “Confidential to Editor” section, and submit your "Accept" recommendation.

Reviewer #2: All comments have been addressed

2. Is the manuscript technically sound, and do the data support the conclusions?

Reviewer #2: Yes

3. Has the statistical analysis been performed appropriately and rigorously? 

Reviewer #2: Yes

4. Have the authors made all data underlying the findings in their manuscript fully available?

Reviewer #2: Yes

5. Is the manuscript presented in an intelligible fashion and written in standard English?

Reviewer #2: Yes

6. Review Comments to the Author

Reviewer #2: (No Response)

7. PLOS authors have the option to publish the peer review history of their article (what does this mean?). If published, this will include your full peer review and any attached files.

Reviewer #2: **Yes: **Dengchuan Wang

---

## [Editor Report · Acceptance letter]

26 Mar 2021

PONE-D-20-34707R1 

Ethical and psychosocial considerations for hospital personnel in the Covid-19 crisis: Moral injury and resilience 

Dear Dr. Kreh:

I'm pleased to inform you that your manuscript has been deemed suitable for publication in PLOS ONE. Congratulations! Your manuscript is now with our production department. 

Kind regards, 

on behalf of

Dr. Francesco Di Gennaro 

Academic Editor

PLOS ONE